# Rapid and Efficient In Vitro Propagation Protocol of Endangered Wild Prickly Pear Growing in Eastern Morocco

Ahmed Marhri [1,*], Aziz Tikent [1], Laurine Garros [2], Othmane Merah [3], Ahmed Elamrani [1], Christophe Hano [2], Malika Abid [1] and Mohamed Addi [1,*]

1 Laboratory for Agricultural Productions Improvement, Biotechnology and Environment (LAPABE), Faculty of Sciences, University Mohammed First, BP-717, Oujda 60000, Morocco
2 Laboratoire de Biologie des Ligneux et des Grandes Cultures, INRAE USC1328, Campus Eure et Loir, Orleans University, 28000 Chartres, France
3 Laboratoire de Chimie Agro-Industrielle (LCA), Université de Toulouse, INRA, INPT, 31030 Toulouse, France
* Correspondence: ahmed.marhri@ump.ac.ma (A.M.); m.addi@ump.ac.ma (M.A.)

**Abstract:** The *Opuntia* genus is widely recognized as a significant member of the Cactaceae family. The eastern Moroccan prickly pear's wild ecotype is renowned for its production of fruits of superior quality, which are in high demand. Nonetheless, the white cochineal (*Dactylopius opuntiae*) has emerged as a significant hazard to the persistence of the indigenous wild prickly pear population in the majority of the country's territories. Conventional plant propagation techniques may facilitate the transmission of pathogens to successive generations and thus fail to satisfy market requirements. Therefore, the primary goal of this study was to develop a rapid and efficient protocol for large-scale production of the eastern Moroccan wild ecotype using axillary buds as the starting material. Sterilization of the starting material is a crucial but challenging step in this species, as the meristem is located just beneath the spine. The protocol developed for this study produced moderately satisfactory results, with 20 to 30% contamination in each experiment. The obtained shoots were incubated on Murashige and Skoog medium supplemented with varying concentrations of BAP (0, 1.5, 3, 5, and 7.5 mg/L). The treatment with BAP at 5 mg/L exhibited a statistically significant increase in the average number of regenerated shoots per explant (19.42). The effect of kinetin on the rooting of prickly pear was evaluated by transferring the shoots to a MS medium supplemented with varying concentrations of kinetin (0, 0.5, 1, 1.5, 2, and 2.5 mg/L). The use of kinetin increased the number and length of roots while also shortening the root development period from 21 days to 10 days. The best results were obtained at a concentration of 1.5 mg/L of kinetin. Furthermore, satisfactory acclimatization of plants was achieved by using plastic containers with a gradually increasing opening of the lids. The outcomes of this experimentation have significant potential to facilitate the preservation of this botanical variety, reduce the risk of white cochineal infestation, and address the need for superior quality fruit supply in the market.

**Keywords:** wild prickly pear; eastern Morocco; in vitro culture; rapid and efficient protocol; large-scale production; BAP; newly regenerated shoots; explant; kinetin; rooting

## 1. Introduction

The nopal cactus, commonly referred to as prickly pear, encompasses various members of the Cactaceae family and is recognized for its distinctive pear-like shape and spines. With over 2000 species included, the genus *Opuntia* is native to the American continent [1], with Mexico boasting the highest number of endemic species, totaling over 600 [2]. Prickly pear has gained worldwide recognition, offering a promising option for fruit and fodder crops, particularly in drought-prone regions. In effect, the prickly pear is characterized by drought resistance and water-use efficiency, as well as adaptability to poor soils. As reported by [3], water stress has no significant impact on the survival of explants; in

contrast, drought stress promotes rhizogenesis and the development of isolated shoots. The plant's notable ability to acclimate and flourish in arid and semi-arid regions, despite the persisting water shortage challenge, can be deemed the key factor behind its prompt and pervasive distribution success [2]. However, the nopal cactus is much more than a resilient plant species that efficiently utilizes water. Its versatility extends to a myriad of uses, including food, cosmetics, pharmaceuticals, and traditional and modern medicine. Furthermore, this adaptable plant plays a critical role in combating desertification and soil erosion, providing an all-encompassing solution to environmental challenges. From agriculture to pharmaceuticals, the nopal cactus is a valuable resource, and its potential for sustainable growth is continually being explored [4,5].

The aforementioned characteristics ultimately lead to the incessant dissemination of this plant throughout the world. In Morocco, starting in 2008, the Green Morocco Plan and National Plan to Combat Desertification have endorsed the promotion of dryland development and the cultivation of crops that can endure the impacts of global warming. Consequently, there has been a remarkable growth in the cactus sector, manifested by an increase in cultivated acreage and national production [6]. In fact, more than 4000 hectares are planted in the central and southern parts of the country each year [7]. Unfortunately, despite all the emergency interventions adopted by local authorities (including the burning of more than 400 hectares of plantations in the Doukkala region) since the first discovery of *Dactylopius opuntiae* at the end of 2014 [8], the cochineal spread rapidly and caused severe damage throughout the country [9]. Even the most popular and highly valued wild variants are presently facing the ominous threat of extinction [10].

In the traditional sense, in vivo propagation involves the rooting of one or more cladodes or small portions of mature cladodes that contain two or more areoles. In the vertical sections of the cladodes, 100% root development was observed, with 80 being the highest number of roots [11]. Although these techniques are effective and easy to perform, their multiplication rates remain low, resulting in a limited number of plants and necessitating large areas for propagation, which is insufficient to meet the growing market demand [2]. Moreover, traditional propagation methods pose a high risk of pest transmission. In contrast, seeds serve as a fundamental unit for diversity and genetic enhancement. Despite the significant role of sexual reproduction in scientific research, which aims to study genetic variability and factors that affect the germination process, the propagation of seeds remains limited to this domain. Seeds do not ensure genetic stability and lead to genetic segregation of offspring, thereby contributing to a prolonged juvenile stage that retards the growth of cacti in general [12].

Whether the aim of plant multiplication is scientific research or commercial purposes, in vitro culture offers a superior solution to the challenges associated with traditional methods. Micropropagation, which enables a high multiplication rate and uniformity of propagated material, can generate millions of new shoots from a single explant without significant mutations or permanent epigenetic changes. As mentioned by [13], micro-propagules of prickly pear are characterized by low genetic variation, with 91% similarity and just 2.79% of the total genetic variation. Additionally, this technique requires less space and produces healthy plants free of pathogens and pests, making it the most desirable option. Furthermore, micropropagation is not limited by season, providing high-quality material throughout the year [14]. Specifically, this study focuses on areole activation, which is one of the most widely used techniques in this field. By stimulating axillary buds, numerous shoots can be produced and repeatedly subcultured, resulting in the regeneration of an extremely high number of shoots. This remarkable feature of micropropagation is one of its most significant advantages [15].

The application of cytokinin can awaken previously dormant axillary buds from the constraints of apical dominance. Moreover, elongating the axillary shoots is considered a simpler approach that maintains genetic stability [16]. Regarding the rooting of shoots, auxins are the most commonly used exogenous phytohormones for the development of the root system, and IBA growth hormone is more effective than NAA alone or in

combination with IBA [17]. In vitro propagation, plants could respond differently, even if they belong to the same species. Additionally, micropropagation is highly dependent on genotype [3]. Thus, the primary goal of this study is to enhance an effective protocol for in vitro mass propagation to aid in the preservation of the endangered wild prickly pear in eastern Morocco.

## 2. Materials and Methods

### 2.1. Explant Sampling and Disinfection

Sampling was conducted in the rural community of Sidi Bouhria, located in eastern Morocco (34°44′15.4″ N 2°21′15.188″ W) (Figure 1). The healthy young cladodes, ranging from 5 to 10 cm in length, were selected as explants after the wild parents were chosen. Approximately ten complete cladodes were carefully cut at their bases from the chosen plant. The cut cladodes were then preserved in an isothermal bag from the moment of excision until they were transported to the laboratory.

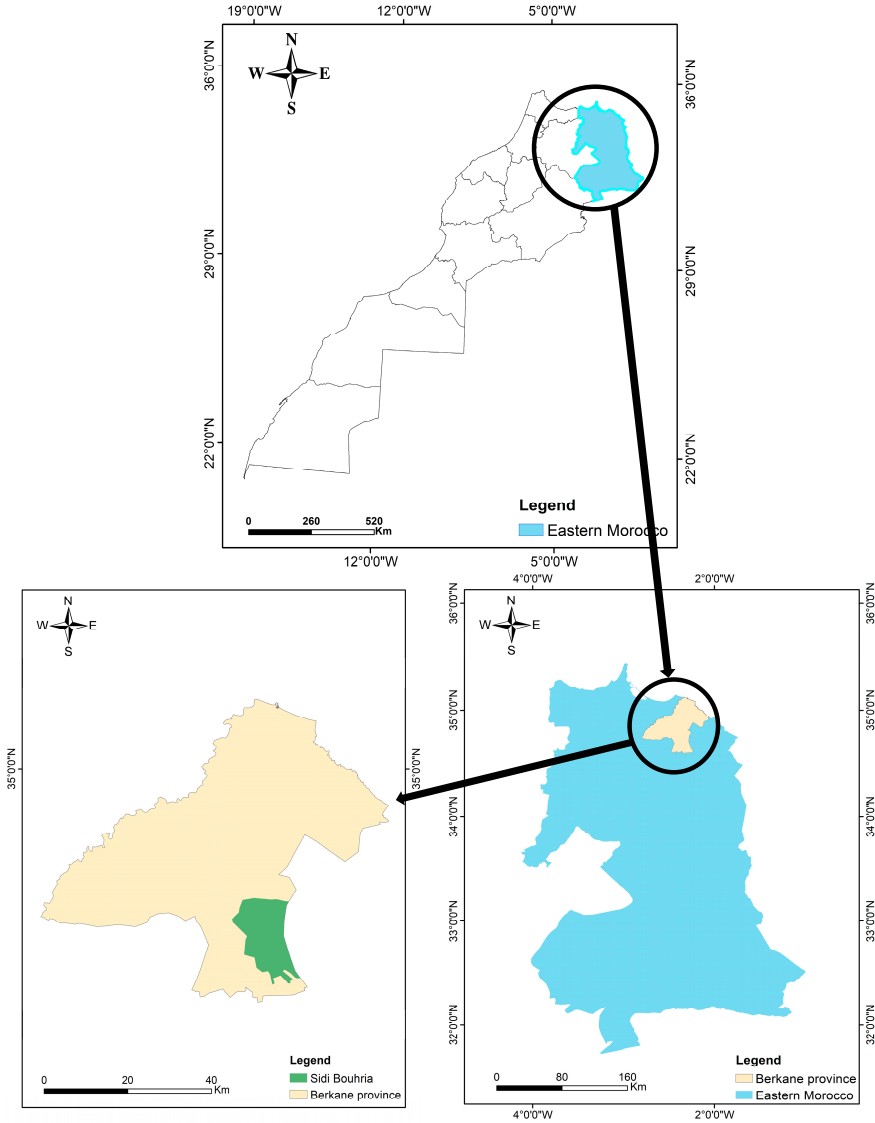

**Figure 1.** Location of the sampling site (Sidi Bouhria eastern region of Morocco).

To ensure that the entire cladodes were clean, the spines were meticulously chopped to a minimum size using a sharp chisel. This was carried out with the utmost care so as not to injure other tissues. The cladodes were then rinsed with tap water. In order to eliminate any surface contamination of the tissues, aseptic in vitro conditions were employed. Immersion

in 70% (*v/v*) ethanol for 1 min was followed by 20 min in a commercial bleach solution (sodium hypochlorite) (15%) containing 5 mL/L Tween 20 and succeeded by three washes of 10 min each with sterile distilled water.

## 2.2. Initiation of Culture and Incubation Conditions

In this stage, all parts of the cladode that contain areoles were utilized as explants. Explants were cut into pieces of about 1 cm$^2$, each containing an areole. The disinfected explants were then aseptically incubated in petri dishes containing 20 mL of half-strength Murashige and Skoog (MS/2) medium supplemented with 25 g/L sucrose. This medium was further supplemented with 100 mg/L myo-inositol, 0.1 mg/L thiamine hydrochloride, 0.25 mg/L nicotinic acid, 0.25 mg/L pyridoxine hydrochloride, and 0.7% agar. This basal medium was then adjusted to a pH of 5.8 before being sterilized in an autoclave (121 °C, one bar) for 20 min. All experiments were conducted under sterile conditions to prevent any potential contamination (Figure 2). Each replicate consists of thirty explants per treatment, and the experiments were repeated at least three times to confirm the results.

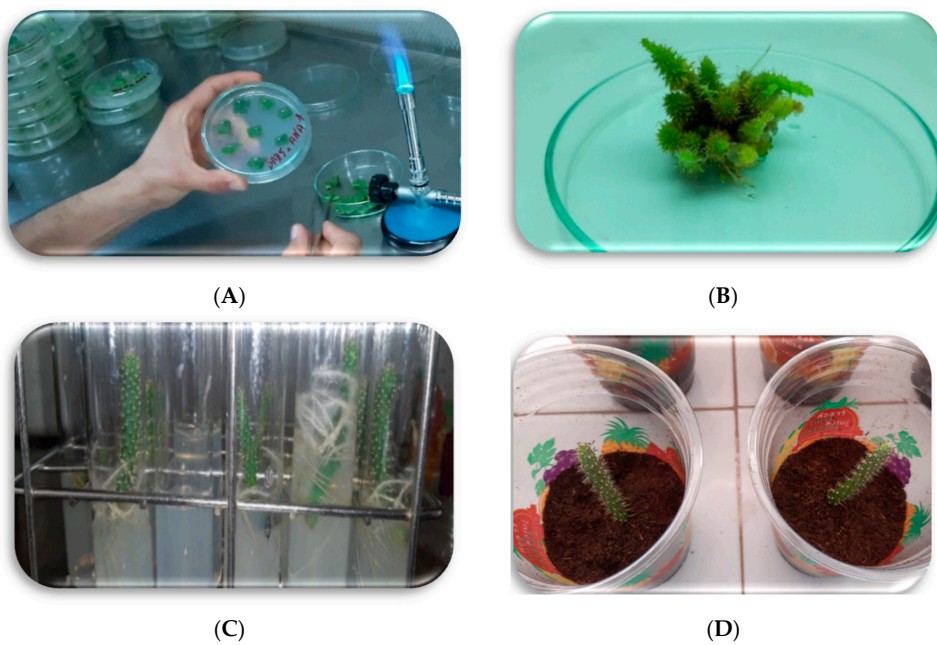

(**A**)  (**B**)

(**C**)  (**D**)

**Figure 2.** Micropropagation of wild prickly pear (**A–D**). (**A**) Areole incubation of one cm$^2$ under sterile conditions. (**B**) Multiple shoot formation from a single explant on MS medium supplemented with BAP 5.0 mg/L. (**C**) Rooting in shoots on MS supplemented with kinetin 1.5 mg/L. (**D**) gradual acclimatization of plantlet.

Finally, the petri dishes were sealed with Parafilm™ (Bemis, PM-996, Neenah, WI, USA) and incubated in a growth chamber with a photoperiod of 16 h of light and 8 h of darkness at 25 ± 2 °C for 8 weeks. The same incubation conditions were utilized in all subsequent experiments to ensure consistency and reliability of the results.

## 2.3. Proliferation Phase

Two months after the initial culture, the activated buds were transferred to a proliferation medium, which was a full Murashige and Skoog (MS) medium supplemented with various concentrations of hormones. These explants (Figure 2) were then subjected to different treatments, including benzylaminopurine (BAP) alone at various dosages (0, 1.5, 3, 5, and 7.5 mg/L) and in combination with naphthaleneacetic acid (NAA) at 2 mg/L. The data were then recorded after eight weeks of culture.

In order to determine the proliferation rate, the number of newly produced shoots per initial shoot (Figure 2) was calculated. The resulting data were analyzed to evaluate the

effectiveness of the different treatments. This information is important for determining the optimal conditions for maximizing shoot proliferation in this particular system.

*2.4. Rooting of Proliferated Shoots*

Additionally, to assess the effect of kinetin on rooting, shoots of 2 cm or more in length were placed on the rooting medium containing 3% sucrose with different concentrations of kinetin (0.5, 1, 1.5, 2, and 2.5 mg/L) and only 0.7% agar. This medium facilitated the removal of rooted shoots and prevented any potential damage to the roots during transplantation of the plantlets into peat.

*2.5. Acclimatization*

Following the removal of plantlets from the culture medium, the roots were thoroughly cleansed using sterile distilled water until all traces of the solid culture medium were eliminated. Subsequently, the plants were gradually acclimatized by repotting them in sterilized peat and covering them with almost fully enclosed plastic boxes for a duration of one week. This was followed by placing them in partially opened pots for another week before they were finally transferred to fully opened pots, as illustrated in Figure 2D. The percentage of successfully acclimated plants was calculated twelve weeks after transplantation.

*2.6. Statistical Analysis*

In the current study, all treatments, whether during the proliferation or the rooted phase, were repeated at least three times, and each replicate consists of thirty explants.

The significance of differences between treatments was evaluated using an analysis of variance (ANOVA) through the IBM SPSS Statistics 25 software. In instances where significant differences ($p \leq 0.05$) were identified, a multiple comparison test of means, specifically the Tukey test, was employed. The data obtained were presented as the mean with one standard deviation.

## 3. Results

The disinfection protocol employed in the present investigation is relatively satisfactory. However, between 20 and 30 percent of the explants exhibited signs of infection or discoloration after each attempt. Concerning the proliferation stage, shoots larger than 1 cm that develop from axillary buds are utilized as secondary explants. Analysis of variance demonstrated significant variations in the means of the four groups, revealing that the concentration of BAP growth hormone had a direct impact on shoot proliferation. BAP alone performed the best at a concentration of 5 mg/L, producing an average of 19.42 shoots, which proved to be the most efficient number of shoots regenerated from a single shoot compared to the other doses tested. Furthermore, BAP 1.5 mg/L and the combination of BAP 5 with NAA 2 mg/L produced a low number of shoots, making them the least effective, with mean values of 2.83 and 3.42, respectively (Table 1). The two BAP treatments, 3 and 7.5 mg/L, yielded more or less satisfactory outcomes with values of 6.92 and 6.17, respectively. No proliferation was observed in the control group since no exogenous growth regulators were utilized.

Our findings demonstrate that two doses of kinetin, 1.5 mg/L and 2 mg/L, exhibited the highest root formation, yielding an average of 9.08 and 8.75 roots, respectively, as illustrated in Table 2. Notably, no significant difference in root formation was observed between the two doses. Conversely, the results of other treatments, administered at doses exceeding 2 mg/L or less than 1.5 mg/L, were found to be less efficient.

**Table 1.** Effect of different hormone combinations in MS media for shoot proliferation.

| Treatments (mg/L) | Shoot Number per Explant |
|---|---|
| Control (hormone free) | 0 [a] |
| BAP 1.5 | 2.83 [b] $\pm$ 1.22 |
| BAP 3 | 6.92 [c] $\pm$ 2.49 |
| BAP 5 | 19.42 [d] $\pm$ 2.52 |
| BAP 7.5 | 6.17 [c] $\pm$ 1.74 |
| BAP 5 + NAA 2 | 3.42 [b] $\pm$ 1.38 |

According to the analysis of variance, means followed by the same letters are not significantly different ($p \leq 0.05$).

**Table 2.** Effect of kinetin on root regeneration.

| Treatments (mg/L) | Root Number per Explant |
|---|---|
| Control | 1.58 [a] $\pm$ 0.62 |
| Kinetin 0.5 | 2.33 [a] $\pm$ 1.03 |
| Kinetin 1 | 5.33 [b] $\pm$ 1.56 |
| Kinetin 1.5 | 9.08 [d] $\pm$ 1.56 |
| Kinetin 2 | 8.75 [d] $\pm$ 1.68 |
| Kinetin 2.5 | 7.00 [c] $\pm$ 1.84 |

According to Tukey's test, means followed by the same letters are not significantly different ($p \leq 0.05$).

Once the plant attains a minimum root length of three centimeters, the acclimation phase is initiated. This final step of the experiment is typically unchallenging, with a success rate of 90 percent, with 36 out of 40 plants adapting successfully within three months.

## 4. Discussion

The initial stage of in vitro culture involves the critical process of disinfection, which necessitates a meticulous decontamination of the explants. This step is considered a limiting factor for in vitro culture via areoles. Firstly, the meristem is situated beneath the spines, rendering it particularly vulnerable to cleaning and disinfecting agents. Secondly, phenolic compounds secreted by the plant may result in necrosis of the explants. For these reasons, several researchers advocate using seeds to generate shoots, which can be utilized as a source of secondary explants, as outlined in [18]. It is, therefore, recommended to avoid using detergents with high concentrations.

The technique of micropropagation has become increasingly popular for implementing multiplication strategies in various cactus species. However, there seems to be insufficient research on using this technique for mass propagation of *Opuntia* species through the proliferation of axillary buds [19]. As per the aforementioned findings, the response of shoot proliferation varied in a cytokinin concentration-dependent manner, as also previously documented by researchers [20,21]. Several studies have aimed to investigate the effect of BAP [22–24]. Our research demonstrates that the most effective shoot multiplication rate was achieved with 5 mg/L BAP treatment, yielding an average of 19.42, consistent with the outcomes of [24], who observed the highest proliferation rate using the same treatment without any combination, with an average of 26.5 regenerated shoots per explant. Our results are inconsistent with those of [25], who found that BAP at 3 mg/L gave the highest number of proliferated shoots (21), while increasing the concentration of this phytohormone (4 mg/L) gave less satisfactory results. According to the obtained results, the combination of BAP with NAA decreases the number of proliferated shoots. However, [26] reported that the combination of 5 mg/L and 0.25 mg/L of BAP and NAA, respectively, gave the best results in the shoot's development. In addition to the use of BAP described in the current study, the growth hormone BAP could also be used in combination with 2,4D to induce callus [27].

In general, auxins are the most commonly used phytohormones to promote adventitious root development [4,22,24,28,29], while cytokinins are used only to a very limited extent for rhizogenesis. Therefore, the effect of cytokinin on rhizogenesis in various species is still unknown. In fact, few studies conducted on other species have reported the remarkable effect of cytokinins such as zeatin, kinetin, and thidiazuron on the rooting of shoots, with kinetin providing the most satisfactory results in terms of root length [30]. In addition, the efficacy of kinetin alone on rhizogenesis has been reported [31,32]. Nevertheless, no research has been conducted on the utilization of kinetin in the root development of the prickly pear. In the current investigation, kinetin was utilized solely to direct organogenesis toward root development. The optimal outcome was achieved with kinetin at a concentration of 1.5 mg/L, which provided an average of 9.08 roots. In addition, the results indicate that the stimulation of root development is not required in this particular species, as the organ developed adequately within three weeks even in the absence of growth hormones. In fact, following plant wounding, regeneration of missing parts is a natural capability, and the agar medium further enhances the production of adventitious roots [33,34]. Furthermore, under in vitro culture conditions, cacti generate an abundance of auxin [35], which enables adventitious roots to develop on hormone-free MS. However, the current investigation has revealed that the presence of kinetin in the culture medium significantly enhances the number of developed roots in a mere ten days, as opposed to three weeks or longer for untreated plants. As such, the concentration of kinetin directly influences root development. Comparable results have been reported by numerous authors [31,36] who have found that kinetin alone, without any auxin combination, affects rooting both in vitro and ex vitro. The current research confirms that kinetin can be considered an effective phytohormone for directing organogenesis towards rhizogenesis, facilitating the full development of roots in all regenerated shoots. These findings prompt further investigations into the effects of kinetin and other cytokinins on the rooting of prickly pear shoots and their comparison to the conventional use of auxins for this purpose.

Generally, plantlets derived from in vitro culture can be easily altered during acclimatization. Indeed, the particular conditions of tissue culture, such as high humidity, confined containers that impede gas exchange, and a high dose of exogenous growth regulators [18], contribute to the development of morphologically and physiologically abnormal plantlets [37]. It is important to consider the effect of cytokinins, such as kinetin, on acclimating micropropagules. In effect, abscisic acid closes stomata, while cytokinin has the opposite effect, resulting in the permanent opening of stomata and causing plantlets to lose water and dry out. Additionally, the conditions in the vessels used for tissue culture, such as high irradiance, are very different from the field. Despite these challenges, cacti are known to have a high success rate in acclimatizing, with a success rate of 100% reported in previous studies [20,23,24]. This success can be attributed to the well-developed root system of Opuntiaceae, which enables high rehydration capacity. Furthermore, a progressive acclimatization method, which involves using a plastic cover for 2–3 weeks to increase external humidity and correct abnormalities, can help avoid desiccation.

## 5. Conclusions

The escalating prominence of cactus research, specifically for the prickly pear species, is primarily attributed to various factors such as climate change, persistent drought, desertification, and the recent threat of *Dactylopius opuntiae*. This study has aimed to develop a reliable and effective micropropagation protocol for this crucial wild species, which could facilitate its conservation. In addition to the swift and copious production of healthy and uniform plants, the current research is also focused on exploring the impact of kinetin on shoot rooting. Our findings have demonstrated that a concentration of 5 mg/L of BAP produced the maximum number of regenerated shoots per explant.

Remarkably, the prickly pear exhibited a remarkable capability of root development, even in the absence of exogenous hormones. However, the use of hormones could significantly hasten this process, and in this study, kinetin was used instead of auxins. The results

showed that the kinetin was effective, and the roots developed in only ten days, compared to the shoots incubated in MS medium without phytohormone, which took three weeks or longer. Therefore, our study establishes a reliable protocol for micropropagation, and the successful use of kinetin in promoting rapid and efficient rooting could have significant implications for the conservation and cultivation of prickly pear.

**Author Contributions:** Conceptualization, A.M., M.A. (Malika Abid) and M.A. (Mohamed Addi); methodology, A.M.; software, A.T.; validation, M.A. (Malika Abid), C.H. and M.A. (Mohamed Addi); formal analysis, L.G.; investigation, A.M.; resources, M.A. (Malika Abid); data curation, O.M.; writing—original draft preparation, A.M.; writing—review and editing, M.A. (Malika Abid) and M.A. (Mohamed Addi); visualization, O.M.; supervision, M.A. (Malika Abid) and M.A. (Mohamed Addi); project administration, M.A. (Mohamed Addi); review and editing: A.E. All authors have read and agreed to the published version of the manuscript.

**Funding:** This research received no external funding.

**Institutional Review Board Statement:** Not applicable.

**Informed Consent Statement:** Not applicable.

**Data Availability Statement:** Not applicable.

**Conflicts of Interest:** The authors declare no conflict of interest.

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
