# Peer review of "Rapid and Efficient In Vitro Propagation Protocol of Endangered Wild Prickly Pear Growing in Eastern Morocco"

_horticulturae, doi:10.3390/horticulturae9040491_

Round 1

Reviewer 1 Report

The paper concerns on Opuntia ficus-indica, which is very important species due to food production, pharmaceuticals or cosmetics. On the other hand this plant have a huge potential to recultivation of arid areas, drylands, because it is very tolerant to drought. There is a global need to multiply plants tolerant to harsh conditions especially arid and dry sites due to global warming and contamination. In vitro micropropagation is a common and very useful tool to get plant lines tolerant to such conditions.

The Introduction section must be improved. You should write more about the micropropagation of Opuntia. You should cite papers like e.g.:

Mohamed-Yasseen, Y., Barringer, S. A., Splittstoesser, W. E., & Schnell, R. J. (1995). Rapid propagation of tuna (Opuntia ficus-indica) and plant establishment in soil. Plant cell, tissue and organ culture42, 117-119.

Shehu, U. I., Sani, L. A., & Ibrahim, A. B. (2016). Auxin induced rooting of cactus pear (Opuntia ficus-indica L. Miller) cladodes for rapid on-farm propagation. African Journal of Agricultural Research11(10), 898-900.

Zoghlami, N., Bouamama, B., Khammassi, M., & Ghorbel, A. (2012). Genetic stability of long-term micropropagated Opuntia ficus-indica (L.) Mill. plantlets as assessed by molecular tools: Perspectives for in vitro conservation. Industrial Crops and Products36(1), 59-64.

The vegetative propagation e.g.: Stambouli-Essassi, S., Harrabi, R., Bouzid, S., & Harzallah-Skhiri, F. (2015). Evaluation of the efficiency of Opuntia ficus-indica cladode cuttings for vegetative multiplication. Notulae Botanicae Horti Agrobotanici Cluj-Napoca43(2), 521-527.

Tolerance: Radi, H., Bouchiha, F., El Maataoui, S. et al. Morphological and physio-biochemical responses of cactus pear (Opuntia ficus indica (L.) Mill.) organogenic cultures to salt and drought stresses induced in vitro. Plant Cell Tiss Organ Cult (2023).

Line 62.: D. opuntiae – write full name

Line 83-85:” Specifically, this study focuses on areole activation, which is one of the most widely used techniques in this field. By stimulating axillary buds, numerous shoots can be produced and repeatedly subcultured, resulting in the regeneration of an extremely high number of shoots” - please, quote citation to this finding

Line 99-102: It is too long and too verbose, change the sentence to: Material for study were collected in Sidi Bouhria, the eastern region of Morocco (34° 44' 15.4" N 2° 21' 15.188" W) at elevation 701 meters (Fig. 1). The healthy young cladodes (5-10 cm in length) were sampled.

Line 99-100: How much samples of cladodes – please provide the exact numbers.

Line 103-112: Please rewrite, it is too long there is no need comments as e.g.: “This process successfully eradicated any microbial contamination.”
E.g. you write: “Finally, the detergents were removed by rinsing the cladodes five times with sterile distilled water. To ensure the complete elimination of excess humidity, the cladodes were dried for half an hour before proceeding with further experimentation.” It could be shorter: Then the explants were rinsed five times with distilled water and dried for ½ h.

Line 112: please provide at what temperature and what equipment was used.

Line 118: “These explants are disinfected and then cut into 1 cm2 pieces, with each piece containing  an areole.” Change into: explants were cut into 1 cm2 pieces containing an areole.

Line 119: “This process is carried out under sterile conditions in a laminar air flow system 119 (Figure 2) to prevent any potential contamination” – throw away, write it at the end e.g. all experiments were conducted on sterile conditions.

Line 95-175: Please rewrite all the Material and methods according to my suggestions above and other scientific papers. The text is too verbose, avoid comments etc.

Line 122: Correct: (MS/2) into (MS)

Line 122-123: “It is important to note that even if the explant is large (more than 0.5 cm3), it is still inoculated.” – I do not understand this sentence, please rewrite.

Line 124-128: I do not understand. Firstly explants were kept on half strength MS and then transferred into this medium? Is it one medium or different media? Please rewrite.

Line 146-149: throw it away. There is no important information. Write shortly and only specifics.

Line 151-153: The same as above.

Line 150-155: Is it was a one medium or the explants firstly was on medium containing 3% sucrose and only 0.7% agar only and then on different media containing kinetin? It should be rewrite.

Line 157-159: I do not understand. Do you used this method for preparing plants to the ex vitro conditions? It should be rewrite or deleted.

Line 161: Instead of shoots it should be plantlets.

Line 170-171: How it was done? What do you mean by replication in this experimental model?

Discussion: please provide more previous researches about the regeneration and micropropagation of Opuntia and prove the novelty of your work. There is so many papers where authors micropropagate Opuntia, so you should clearly indicate the validity of your work in the discussion section. You write that other authors also use kinetin to root formation, so what is novel in your work?

You have to refer to e.g.:

Bouzroud, S., El Maaiden, E., Sobeh, M., Devkota, K. P., Boukcim, H., Kouisni, L., & El Kharrassi, Y. (2022). Micropropagation of Opuntia and Other Cacti Species Through Axillary Shoot Proliferation: A Comprehensive Review. Frontiers in Plant Science13, 926653.

Ghaffari, A., Hasanloo, T., & Nekouei, M. K. (2013). Micropropagation of tuna (Opuntia ficus–indica) and effect of medium composition on proliferation and rooting. Int J Biosci3, 129-139.

Bouchiha, F., & Mazri, M. A. (2022). Micropropagation of cactus pear (Opuntia ficus indica) by organogenesis. African and Mediterranean Agricultural Journal-Al Awamia, (135), 1-15.

Angulo-Bejarano, P. I., & Paredes-López, O. (2011). Development of a regeneration protocol through indirect organogenesis in prickly pear cactus (Opuntia ficus-indica (L.) Mill). Scientia Horticulturae128(3), 283-288.

Rodríguez-de la OJL, R. P. (2020). PE. Micropropagation of selected materials of Opuntia ficus indica l through culture in vitro of areols. J Appl Biotechnol Bioeng7(3), 121-126.

Mohamed-Yasseen, Y., Barringer, S. A., Splittstoesser, W. E., & Schnell, R. J. (1995). Rapid propagation of tuna (Opuntia ficus-indica) and plant establishment in soil. Plant cell, tissue and organ culture42, 117-119.

Line 285: no superscript

Author Response

Responses to Reviewer 1

Dear Reviewer,

Thank you for giving us the opportunity to improve our manuscript by the revised version and thank to your useful comments.

We really appreciate Reviewers’ comments. We have carefully considered all the feedback and have incorporated the changes suggested by the reviewer, which are highlighted in green in the manuscript. Our responses to the reviewer's comments are provided below, with the comments in bold.

We hope this revision will satisfy reviewers queries, and that our work will be considered for publication in Horticulturae.

With kind regards

Dr Addi and the co-Authors

The Introduction section must be improved. You should write more about the micropropagation of Opuntia. You should cite papers like e.g.:

Mohamed-Yasseen, Y., Barringer, S. A., Splittstoesser, W. E., & Schnell, R. J. (1995). Rapid propagation of tuna (Opuntia ficus-indica) and plant establishment in soil. Plant cell, tissue and organ culture, 42, 117-119.

Shehu, U. I., Sani, L. A., & Ibrahim, A. B. (2016). Auxin induced rooting of cactus pear (Opuntia ficus-indica L. Miller) cladodes for rapid on-farm propagation. African Journal of Agricultural Research, 11(10), 898-900.

Zoghlami, N., Bouamama, B., Khammassi, M., & Ghorbel, A. (2012). Genetic stability of long-term micropropagated Opuntia ficus-indica (L.) Mill. plantlets as assessed by molecular tools: Perspectives for in vitro conservation. Industrial Crops and Products, 36(1), 59-64.

The vegetative propagation e.g.: Stambouli-Essassi, S., Harrabi, R., Bouzid, S., & Harzallah-Skhiri, F. (2015). Evaluation of the efficiency of Opuntia ficus-indica cladode cuttings for vegetative multiplication. Notulae Botanicae Horti Agrobotanici Cluj-Napoca, 43(2), 521-527.

Tolerance: Radi, H., Bouchiha, F., El Maataoui, S. et al. Morphological and physio-biochemical responses of cactus pear (Opuntia ficus indica (L.) Mill.) organogenic cultures to salt and drought stresses induced in vitro. Plant Cell Tiss Organ Cult (2023).

Reply: Thank you so much for your important comment. The recommended correction has been made

Line 62.: D. opuntiae – write full name

Reply: The species name is written according to the recommendations

Line 83-85:” Specifically, this study focuses on areole activation, which is one of the most widely used techniques in this field. By stimulating axillary buds, numerous shoots can be produced and repeatedly subcultured, resulting in the regeneration of an extremely high number of shoots” - please, quote citation to this finding

Reply: Thank you very much for your valuable suggestion. The recommended correction has been made.

Line 99-102: It is too long and too verbose, change the sentence to: Material for study were collected in Sidi Bouhria, the eastern region of Morocco (34° 44' 15.4" N 2° 21' 15.188" W) at elevation 701 meters (Fig. 1). The healthy young cladodes (5-10 cm in length) were sampled.

Reply: Thank you so much for your comment. The suggested changes were made.

Line 99-100: How much samples of cladodes – please provide the exact numbers.

Reply: The recommended correction has been made.

Line 103-112: Please rewrite, it is too long there is no need comments as e.g.: “This process successfully eradicated any microbial contamination.”

Reply: Thank you very much. The recommended correction has been made

E.g. you write: “Finally, the detergents were removed by rinsing the cladodes five times with sterile distilled water. To ensure the complete elimination of excess humidity, the cladodes were dried for half an hour before proceeding with further experimentation.” It could be shorter: Then the explants were rinsed five times with distilled water and dried for ½ h.

Reply: Thank you for this important note. The suggested change has been made.

Line 118: “These explants are disinfected and then cut into 1 cm2 pieces, with each piece containing  an areole.” Change into: explants were cut into 1 cm2 pieces containing an areole.

Reply: Thank you so much for your important comment. The recommended change has been made.

Line 119: “This process is carried out under sterile conditions in a laminar air flow system 119 (Figure 2) to prevent any potential contamination” – throw away, write it at the end e.g. all experiments were conducted on sterile conditions.

Reply: Thank you so much for your important suggestion. The adjustment has been implemented as per the suggestion.

Line 95-175: Please rewrite all the Material and methods according to my suggestions above and other scientific papers. The text is too verbose, avoid comments etc.

Reply: Thank you so much for the important comment. The recommended adjustment has been executed

Line 122: Correct: (MS/2) into (MS)

Reply: The recommended correction has been made.

Line 122-123: “It is important to note that even if the explant is large (more than 0.5 cm3), it is still inoculated.” – I do not understand this sentence, please rewrite.

Reply: we appreciate your valuable feedback, thank you. We have made the recommended corrections

Line 124-128: I do not understand. Firstly explants were kept on half strength MS and then transferred into this medium? Is it one medium or different media? Please rewrite. done

Reply: Thank you for the insightful comment. The recommended rectification has been applied.

Line 146-149: throw it away. There is no important information. Write shortly and only specifics.

Reply: thank you so much for the highly valued input. The adjustment has been implemented as per the suggestion

Line 151-153: The same as above.

Reply: The recommended correction has been made.

Line 150-155: Is it was a one medium or the explants firstly was on medium containing 3% sucrose and only 0.7% agar only and then on different media containing kinetin? It should be rewrite.

Reply: Thank you for the insightful suggestion. The adjustment has been implemented

Line 157-159: I do not understand. Do you used this method for preparing plants to the ex-vitro conditions? It should be rewrite or deleted.

Reply: Thank you so much for the important comment. The recommended change has been made.

Line 161: Instead of shoots it should be plantlets.

Reply: The recommended correction has been made.

Line 170-171: How it was done? What do you mean by replication in this experimental model? Done: your right sir the word is not correct. Repeated is the right word

Reply: thank you so much for the highly valued input. We have made the recommended corrections

Discussion: please provide more previous researches about the regeneration and micropropagation of Opuntia and prove the novelty of your work. There is so many papers where authors micro propagate opuntia, so you should clearly indicate the validity of your work in the discussion section. You write that other authors also use kinetin to root formation, so what is novel in your work?

You have to refer to e.g.:

Bouzroud, S., El Maaiden, E., Sobeh, M., Devkota, K. P., Boukcim, H., Kouisni, L., & El Kharrassi, Y. (2022). Micropropagation of Opuntia and Other Cacti Species Through Axillary Shoot Proliferation: A Comprehensive Review. Frontiers in Plant Science13, 926653.

Ghaffari, A., Hasanloo, T., & Nekouei, M. K. (2013). Micropropagation of tuna (Opuntia ficus–indica) and effect of medium composition on proliferation and rooting. Int J Biosci3, 129-139.

Bouchiha, F., & Mazri, M. A. (2022). Micropropagation of cactus pear (Opuntia ficus indica) by organogenesis. African and Mediterranean Agricultural Journal-Al Awamia, (135), 1-15.

Angulo-Bejarano, P. I., & Paredes-López, O. (2011). Development of a regeneration protocol through indirect organogenesis in prickly pear cactus (Opuntia ficus-indica (L.) Mill). Scientia Horticulturae128(3), 283-288.

Rodríguez-de la OJL, R. P. (2020). PE. Micropropagation of selected materials of Opuntia ficus indica l through culture in vitro of areols. J Appl Biotechnol Bioeng7(3), 121-126.

Mohamed-Yasseen, Y., Barringer, S. A., Splittstoesser, W. E., & Schnell, R. J. (1995). Rapid propagation of tuna (Opuntia ficus-indica) and plant establishment in soil. Plant cell, tissue and organ culture42, 117-119.

Reply: Thank you for providing this insightful suggestion. The recommended modification has been implemented.

Reviewer 2 Report

Dear Authors,

I have reviewed your manuscript "Rapid and efficient in vitro propagation protocol of endangered wild prickly pear (Opuntia ficus-indica) grown in Eastern Morocco", submitted for publication in Horticulturae.

I appreciate the work that you have put into developing a new protocol for the micropropagation of the prickly pear cactus. However, I have several important remarks regarding how your paper is framed. I want to ask you to consider them seriously, so that your research can be put into a shape suitable for publication.

My most serious remark concerns the comparison of your work to similar published works. Since your work is of relatively low novelty (a considerable number of papers have already been published, which contain successful protocols for the micropropagation of prickly pear), you needed to emphasize what makes it stand out from the previous publications. You chose to put an emphasis on the use of kinetin for rooting, which is indeed relatively novel: although kinetin is sometimes used for the induction of rooting, this has not been reported for prickly pear yet. However, the way are justifying the importance of kinetin in your study, is incorrectly framed. You insist that the use of kinetin enabled to speed up the rooting process; however, you did not compare kinetin to IBA or other auxins within your own study, but instead, you compared your results for kinetin with the results that other research groups reported when they used auxin for rooting. This cannot be taken as a proof that kinetin speeds up the process of rooting, because no one knows how fast your plants would have produced roots if you used auxin in the specific conditions within your lab, for the specific Eastern Moroccan variety of the prickly pear cactus, which you worked on. For that reason, the comparison of your results for rooting on kinetin, with other results published on auxin, is invalid. The most logical thing to do, would have been to use both hormones in your research, to make a parallel comparison between the two. If you are not willing to repeat your experiments in order to make such a comparison, you should give up making comparisons between the two, and stick just to discussing the usefulness of your own protocol. However, a concise rationale as for, why you decided to use kinetin for rooting instead of IBA which is the most routinely used "rooting hormone", should be provided in one sentence, at some point in the Discussion of your paper. Please corroborate this with arguments, and not the way it is currently written (line 233-234).

·         Other remarks:

·         Materials and Methods:

o    The scale of the experiment is not stated. How many samples, approximately, were taken from the nature? How many cladode cubes were sterilized and incubated for shoot regeneration? How many shoots were set for rooting? How many rooted plantlets were set for acclimatization? All these information are missing from the M&M section.

o    line 101: please delete "with utmost care" and instead provide some more informative details, such as: Did you excise entire cladodes, or parts of cladodes? One cladode per plant, or several?

o    line 102: How did you preserve the cladodes from the moment of excision from the mother plant, until their sterilization in your lab? Did you wrap them in a cloth to preserve moist during transport, or keep them in some solution, did you cool them or just keep them at the ambient temperature? All these information are important to anyone who would want to replicate your procedure for their own use.

o    line 111-112: How exactly did you dry the cladodes for half an hour? Did you air-dry them? Did you do that in sterile conditions under laminar flow?

o    line 129: Please specify: "Parafilm™ (Bemis, Neenah, WI, USA)", instead of just "parafilm"

o    Section 2.4 (Rooting): Unlike the other subsections of Materials & Methods, this subsection is particularly poorly written, as if the person who wrote it did not understand the concept and the point of the rooting procedure. This subsection should be thoroughly revised by a senior scientist. Examples:

§  the plants were rooted without a need for acclimation (!?!)

§  the rooted plantlets were removed from the medium and transferred into peat; "next, the shoots were exposed to different concentrations of kinetin" (!!!!!)

§  the last paragraph is also completely nonsensical

o    Section 2.6: Which software was used for statistical analyses?

·         Results: The Results section needs to be substantially shortened, and thoroughly cleaned from the abundant discussion-like parts of the text. Due to the modest amount of numerical results, the Authors might consider removing the subsections, and keeping the entire Results as one, undivided, relatively short section of the article. This would be justified since this is not a full-length research article anyway, but a Communication.

o    lines 178-185: These are not results, this is discussion. Please either delete, or transfer into the Discussion section.

o    lines 189-194: I cannot say if this is methodology or discussion, but it is definitely not a text of the Results. This part should best be deleted.

o    line 209: "two doses of kinetin", not "both doses". You also used other doses apart of these two.

o    line 213: please replace "less significant" with "less efficient"

o    line 216: Did you measure the differences in acclimatization rates between the plants that were rooted on different concentrations of kinetin? Or just the total acclimatization rate of all the rooted plants that you obtained? Please provide the exact number instead of "typically greater than 90%", because this is a result of your work as well, and the numeric results of this work of yours are already very scarce.

·         Discussion: As already stated, the Discussion should be rewritten so as not to rely on the (currently abundant) comparisons between your results for the effects of kinetin on rooting, and the results of previous researchers on the effects of IBA (lines 268-271, 298-299)

o    line 233-234: "The rooting nutrient medium must contain an exogenous supply of kinetin rather than auxin." I disagree. What makes you say that? If you had particular reasons to use kinetin for rooting and not auxin, you should provide valid arguments to support that.

o    line 253: the letter K is missing from kinetin

o    line 261: Is AIB actually IBA??

o    line 280: Low air humidity? In tissue culture? Are you sure?!??

·         Authorship Contribution Statement: Please remove the template text and the quote marks around the Authorship Contribution Statement. Same for the Data Availability Statement.

·         Language and terminology: a thorough revision of language and terminology should be done to ensure accurate formulations throughout the manuscript:

o    cultivar vs. variety – a wild variety of a plant species cannot be called a cultivar. Cultivar is, by definition, a cultivated variety. Please replace the word "cultivar" with "variety" throughout the manuscript text, including the Abstract. Accordingly, please also replace "grown in Eastern Morocco" with "growING in Eastern Morocco" in the title of your article, since "grown" implies domestication.

o    the name of the variety – reading your paper I get the impression that the Eastern Moroccan variety for which you described the protocol, is an important variety of Opuntia for the biodiversity of Morocco. Does it have a particular name – either a Latin name or a local, folk name – that distinguishes it from other varieties of O. ficus-indica? If it does, you should use that name to identify it instead of "Eastern Moroccan variety". Has it been previously described in literature? If yes, please provide a reference within the Introduction. If not, you should briefly state that this variety has not been described in literature previously, and provide the argumentation that corroborates it as a distinct variety.

o    inoculation vs. incubation – the word "inoculation" is used for infectious agents. Please replace "inoculation" and "inoculate" with "incubation" and "incubate" throughout the manuscript text, including the Abstract.

o    the use of Saxon genitive: Please avoid the use of Saxon genitive to designate the belonging to objects ("market's demand", "explants sampling", "explants disinfection", "shoots regeneration", "roots regeneration", etc). Please remember that you are doing research in plant biotechnology, not "plant's biotechnology", "plants biotechnology", nor "plants' biotechnology", although you are not working with a single plant, but with many of them. Accordingly, you should say "market demand", "explant sampling and disinfection", "shoot proliferation", "root regeneration" (lines 19, 35, 96, 177, 188, 219).

o    line 42: Spiny spines??

o    line 43: please replace "this plant" with "the genus Opuntia"

o    line 56-57: please thoroughly rephrase, especially "this civilization"

o    line 62: please fully spell out the genus name at the first mention within the main manuscript text

o    line 70: did you mean pest transmission?

o    line 73: please replace "this technique" with "the propagation through seeds".

o    line 81: please add "and pests" ("pathogens and pests"). Because the pests are a bigger problem here, than pathogens are.

o    Section 2.2: Please put the text of the entire subsection 2.2 into past tense.

Author Response

Responses to Reviewer 2

Dear Reviewer,

Thank you for giving us the opportunity to improve our manuscript by the revised version and thank to your useful comments.

We really appreciate Reviewers’ comments. We have carefully considered all the feedback and have incorporated the changes suggested by the reviewer, which are highlighted in green in the manuscript. Our responses to the reviewer's comments are provided below, with the comments in bold.

We hope this revision will satisfy reviewers queries, and that our work will be considered for publication in Horticulturae.

With kind regards

Dr Addi and the co-Authors

The most serious remark concerns the comparison of the work to similar published works:

Reply: Thank you for your remark. Having implemented the recommended correction, we have chosen to abstain from drawing any further comparisons between our results and the prior findings on auxin that have been published.

Materials and Methods

Reply: The suggested change has been made

Line 101-102

Reply: Thank you very much for your valuable suggestion. The recommended correction has been made.

Line 111-112-129

Reply: Thank you so much for your comment. The suggested changes were made.

Section 2.4 (Rooting)

Reply: Thank you so much. The recommended correction has been made

The plants were rooted without a need for acclimation

The suggested change has been made: deleted sentence

Concerning the sentence: This medium facilitates the removal of rooted shoots and avoids any potential root damage when transplanting seedlings into peat: which means, the new medium, which contains little agar, allows easy and damage-free expulsion of the seedlings during the transition to the acclimatization phase.

The last paragraph is also completely nonsensical

The suggested change has been made: deleted paragraph

Section 2.6. Which software was used for statistical analyses?

Reply: The recommended correction has been made

Results

Lines 178-185- 189-194-209-213

Reply: The recommended corrections have been made

line 216: Did you measure the differences in acclimatization rates between the plants that were rooted on different concentrations of kinetin? Or just the total acclimatization rate of all the rooted plants that you obtained?

Reply: Thank you so much for your important comment. Just the total acclimatization rate of all the rooted plants was calculated

Discussion

As already stated, the Discussion should be rewritten so as not to rely on the (currently abundant) comparisons between your results for the effects of kinetin on rooting, and the results of previous researchers on the effects of IBA (lines 268-271, 298-299)

Thank you very much for your valuable suggestion, the recommended corrections have been made

The comparisons between our results and those found in literature was deleted and replaced by the discuss of our results and the usefulness of our protocol.

Line 233-234- 253-261-280.

Reply: The recommended correction has been made

Round 2

Reviewer 1 Report

Dear Authors,
The manuscript reads much better. I suggest minor revision included below:

Dear Line 32:  mg/L, change into mg/l

Page 5: please change the picture D of fig. 2. It should not be “stretched”.

Line 143: not italic

Line 169: Statistical analysis – what do you mean by repetition/replication? 3 plantlets? Or 3 groups of plantlets? 3 petri dishes? Please provide the specifics/model of the experiment, what was the repetition.

Best wishes

Author Response

Responses to Reviewer 1 round 2

Coauthors and I are grateful for the supportive and helpful feedback provided by the reviewer regarding our manuscript. The comments were comprehensive and greatly improved the quality of the manuscript. We are confident that incorporating the suggestions has improved the scientific merit of the revised manuscript. We have made the recommended changes and highlighted them in blue within the manuscript.

----------------------------------------------------------------------------------------------------------------

Line 32:  mg/L, change into mg/l

Reply: The recommended correction has been made

Page 5: please change the picture D of fig. 2. It should not be “stretched”.

Reply: Thank you very much for your valuable suggestion. The recommended correction has been made.

Line 143: not italic

Reply: The suggested change has been made

Line 169: Statistical analysis – what do you mean by repetition/replication? 3 plantlets? Or 3 groups of plantlets? 3 Petri dishes? Please provide the specifics/model of the experiment, what was the repetition.

Reply: Thank you so much for your comment. The suggested changes were made.

Reviewer 2 Report

Dear Authors,

I have been asked for the second round of review of your manuscript "Rapid and efficient in vitro propagation protocol of endangered wild prickly pear growing in Eastern Morocco", submitted for publication in Horticulturae.

I was happy to see considerable improvements in your manuscript compared to the initial submission. However, your article still requires minor corrections, concerning mostly the addition of a few clarifications, and removal of redundant portions of text, which currently present an unnecessary burden. Also, please make sure that you stick to the description of your original methodology, without altering it to please the reviewers. Please follow my remarks as given below:

·         Abstract:

o   line 18-19: please delete "scientifically known as". The text should read: "Nonetheless, the white cochineal (Dactylopius opuntiae)..."

o   line 31: "concentrations of of kinetin"

o   line 35-37: The penultimate sentence of the Abstract is completely redundant with what has already been said, please delete it.

·         Introduction:

o   Although the Introduction contains the information about prickly pear's North American (Mexican) origin, the subsequent mentioning of wild Moroccan ecotypes leaves the responsibility of filling in the missing dots to the reader. Reading the paragraph 62-70, I felt like a commentary was missing about O. ficus-indica being an invasive species in Morocco, whose invasion however brought more benefit than harm to the local environment (is that even true? A reference to strengthen this argument would be beneficial), due to its beneficial effects in combating soil erosion and desertification.

o   On a related note, the claim that D. opuntiae presents an extinction threat to O. ficus-indica in Morocco (line 70) requires a reference to support the claim. I propose Bouharroud et al. 2018: https://doi.org/10.1111/epp.12471 but the Authors might find that some other reference is more appropriate to support the claim.

o   On another related note, please replace the word "indigenous" with "wild" (line 68-69), because O. ficus-indica is not indigenous in Morocco.

o   line 50-52: The sentence citing the reference no.3 looks like it's been transplanted here for no reason. The in vitro tolerance to drought should not be brought up to corroborate the drought tolerance of a species that is widely recognized as drought tolerant.

o   line 60: please add the letter "D" ("explored").

·         Material & Methods:

o   line 122: The text here looks truncated. There is no full stop at the end of the sentence, and the information about the cladodes being thoroughly rinsed with sterile distilled (or deionized?) water after the bleach/Tween 20 treatment, is missing.

o   line 129: What is "strength Murashige and Skoog"? I believe the word "half-strength" (MS/2) is missing here. Was this the MS, or the MS/2 medium? Reading your response to Reviewer 1, I noticed that Reviewer 1 told you to change "MS/2" to "MS". If you used the MS/2 medium and not full MS for the establishment of the initial culture of prickly pear, you should state the correct medium that you used, not a medium that one of the reviewers thinks that you should have used. Please stick to your original methodological procedure and ignore the suggestions to falsely alter its steps just to please one of the reviewers.

o   line 130-131: This sentence is incorrectly positioned in the text. It should be stated somewhere within this section, but not at this place because it disrupts the description of the procedure. Please displace it, preferably to the beginning of the section 2.2.

o   line 139: Please delete "it is important to note that" from the beginning of the sentence.

·         Results:

o   line 186: "resulted in a reduced number of shoots" – please revise. These growth regulators did not result in a "reduced number of shoots", since their addition to the medium helped shoot regeneration, which did not happen on media without growth regulators. These concentrations were just less efficient than the optimal one.

o   line 187: please delete "refer to".

o   line 194: please delete the remaining subsection title.

o   line 202: 36 out of 40 is not "greater than 90 percent", it exactly equals 90%.

o   line 206: According to Tukey's test, not according to the analysis of variance.

·         Discussion:

o   line 212: I believe that you meant "rendering it", not rendering them particularly vulnerable, because the meristem is vulnerable, not the spines.

o   line 218-219: Please revise into: "there seems to be insufficient research".

o   line 234: "to regenerate calls" – please revise. "Callus" (or "calli" in plural) is misspelled, and besides, calli are not "regenerated" since they are not plant organs such as shoots or roots. Calli are induced, formed, or established (please consult Ikeuchi et al. 2013: https://doi.org/10.1105/tpc.113.116053)

o   line 257-260: These two sentences (the one that cites ref.35 and the next one) are totally not needed in the text. Also, the sentence in line 260-261 may be deleted and the reference to the work by Beck and Caponetti may be simply merged into the sentence that is currently in the lines 255-257, since it is another reference that supports the same claim.

·         References:

o   Ref.38 – the name Pospíšilová is misspelled on the journal publisher's website as Pospóšilová. If you download the paper you will see that the correct spelling is "Pospíšilová".

Author Response

Responses to Reviewer 2 Round 2

Coauthors and I are grateful for the supportive and helpful feedback provided by the reviewer regarding our manuscript. The comments were comprehensive and greatly improved the quality of the manuscript. We are confident that incorporating the suggestions has improved the scientific merit of the revised manuscript. We have made the recommended changes and highlighted them in blue within the manuscript.

--------------------------------------------------------------------------------------------------------------

   Abstract:

line 18-19: please delete "scientifically known as". The text should read: "Nonetheless, the white cochineal (Dactylopius opuntiae)..."

Reply: Thank you so much for your important comment. The recommanded correction has been made

line 31: "concentrations of of kinetin"

Reply: The recommended correction has been made.

line 35-37: The penultimate sentence of the Abstract is completely redundant with what has already been said, please delete it.

Reply: Thank you very much for your valuable suggestion. The recommended correction has been made.

Introduction:

Although the Introduction contains the information about prickly pear's North American (Mexican) origin, the subsequent mentioning of wild Moroccan ecotypes leaves the responsibility of filling in the missing dots to the reader. Reading the paragraph 62-70, I felt like a commentary was missing about O. ficus-indica being an invasive species in Morocco, whose invasion however brought more benefit than harm to the local environment (is that even true? A reference to strengthen this argument would be beneficial), due to its beneficial effects in combating soil erosion and desertification.

Reply: Thank you for this important note. The suggested change has been made.

On a related note, the claim that D. opuntiae presents an extinction threat to O. ficus-indica in Morocco (line 70) requires a reference to support the claim. I propose Bouharroud et al. 2018: https://doi.org/10.1111/epp.12471 but the Authors might find that some other reference is more appropriate to support the claim.

Reply: The recommended correction has been made.

On another related note, please replace the word "indigenous" with "wild" (line 68-69), because O. ficus-indica is not indigenous in Morocco.

Reply: Thank you so much for your important suggestion. The adjustment has been implemented as per the suggestion.

line 50-52: The sentence citing the reference no.3 looks like it's been transplanted here for no reason. The in vitro tolerance to drought should not be brought up to corroborate the drought tolerance of a species that is widely recognized as drought tolerant.

Reply: we appreciate your valuable feedback, thank you. We have made the recommended corrections

line 60: please add the letter "D" ("explored").

Reply: The recommended correction has been made.

Material & Methods:

line 122: The text here looks truncated. There is no full stop at the end of the sentence, and the information about the cladodes being thoroughly rinsed with sterile distilled (or deionized?) water after the bleach/Tween 20 treatment, is missing.

Reply: Thank you for the insightful suggestion. The recommended adjustment has been executed

line 129: What is "strength Murashige and Skoog"? I believe the word "half-strength" (MS/2) is missing here. Was this the MS, or the MS/2 medium? Reading your response to Reviewer 1, I noticed that Reviewer 1 told you to change "MS/2" to "MS". If you used the MS/2 medium and not full MS for the establishment of the initial culture of prickly pear, you should state the correct medium that you used, not a medium that one of the reviewers thinks that you should have used. Please stick to your original methodological procedure and ignore the suggestions to falsely alter its steps just to please one of the reviewers.

Reply: Thank you so much for your important comment. The recommended correction has been made

line 130-131: This sentence is incorrectly positioned in the text. It should be stated somewhere within this section, but not at this place because it disrupts the description of the procedure. Please displace it, preferably to the beginning of the section 2.2.

Reply: thank you so much for the highly valued input. We have made the recommended corrections

line 139: Please delete "it is important to note that" from the beginning of the sentence.

Reply: Thank you so much for your comment. The suggested changes were made.

Results:

line 186: "resulted in a reduced number of shoots" – please revise. These growth regulators did not result in a "reduced number of shoots", since their addition to the medium helped shoot regeneration, which did not happen on media without growth regulators. These concentrations were just less efficient than the optimal one.

Reply: The recommended correction has been made.

line 187: please delete "refer to".

Reply: Thank you very much. The recommended correction has been made

line 194: please delete the remaining subsection title.

Reply: The suggested correction has been made.

line 202: 36 out of 40 is not "greater than 90 percent", it exactly equals 90%.

Reply: Thank you for the important note. The suggested change has been made.

line 206: According to Tukey's test, not according to the analysis of variance.

Reply: Thank you for the insightful suggestion. The adjustment has been implemented

Discussion:

line 212: I believe that you meant "rendering it", not rendering them particularly vulnerable, because the meristem is vulnerable, not the spines.

Reply: we appreciate your valuable feedback, thank you. We have made the recommended corrections

line 218-219: Please revise into: "there seems to be insufficient research".

Reply: Thank you for the insightful suggestion. The recommended adjustment has been executed

line 234: "to regenerate calls" – please revise. "Callus" (or "calli" in plural) is misspelled, and besides, calli are not "regenerated" since they are not plant organs such as shoots or roots. Calli are induced, formed, or established (please consult Ikeuchi et al. 2013: https://doi.org/10.1105/tpc.113.116053)

Reply: The recommended correction has been made.

line 257-260: These two sentences (the one that cites ref.35 and the next one) are totally not needed in the text. Also, the sentence in line 260-261 may be deleted and the reference to the work by Beck and Caponetti may be simply merged into the sentence that is currently in the lines 255-257, since it is another reference that supports the same claim.

Reply: Thank you so much for your important comment. The recommended correction has been made

References:

Ref.38 – the name Pospíšilová is misspelled on the journal publisher's website as Pospóšilová. If you download the paper you will see that the correct spelling is "Pospíšilová".

Reply: Thank you very much for your valuable suggestion. The recommended correction has been made.
